



# Explorative Analysis of Long Time Series of Very High Resolution Spatial Rainfall

Emma Dybro Thomassen[1], Hjalte Jomo Danielsen Sørup[1], Marc Scheibel[2], Thomas Einfalt[3], Karsten Arnbjerg-Nielsen[1]

[1]Department of Environmental Engineering, Technical University of Denmark, Lyngby, 2800, Denmark
[2]Wupperverband, Wuppertal, 42289, Germany
[3]hydro & meteo GmbH&Co.KG, Lübeck, 23552, Germany

*Correspondence to*: Emma Dybro Thomassen (edth@env.dtu.dk)

**Abstract.** Rainfall is often represented by a design storm with uniform intensity in urban hydrological models even though rainfall is a highly dynamic process across very small temporal and spatial scales. This study examines characteristics of high-resolution radar data (5-minute temporal resolution, 1x1 km spatial resolution) over an area of 1824 km$^2$ covering the catchment of the river Wupper, North Rhine-Westphalia, Germany. Extreme events were sampled by a Peak Over Threshold method using several sampling strategies, all based on selecting an average of three events per year. A simple identification- and tracking algorithm for rain cells based on intensity threshold and fitting of ellipsoids, is developed for the study. Both hourly and daily extremes were analysed with respect to a set of 16 descriptive variables. The spatio-temporal properties of the extreme events are explored by means of principal component analysis, cluster analysis, and linear models for these 16 variables. The PCA indicated between 5 and 9 dimensions in the extreme event characteristic data. The cluster analyses identified four rainfall types: extreme convective, convective, convective events in front systems and front system events. The stepwise regression for each variable identified independent variables that correspond well with the correlation structure identified in the clusters. This indicates that the correlation structure may prove useful in setting up a weather generator.

## 1 Introduction

Urban hydrological models of high quality are a required tool to make cities more resilient to pluvial flooding and pollution management. A key input parameter when modelling urban drainage systems is rainfall (Berndtsson and Niemczynowicz, 1988; Schilling, 1991; Thorndahl et al., 2008; Vaes et al., 2001). A common way is to use a model including a rainfall-runoff component that uses rainfall input as either a long-term rainfall series or a design storm (Butler and Davies, 2011; Willems et al., 2012). For some applications rainfall data must be of high spatial and temporal resolution (Berndtsson and Niemczynowicz, 1988; Einfalt et al., 2004; Ochoa-Rodriguez et al., 2015; Schilling, 1991). Schilling (1991) and Einfalt et al. (2004) have proposed resolution requirements of 1-5 minute temporal resolution and 1x1 kilometre spatial resolution.



Inference on properties of rainfall can be based upon two types of data: rain gauge and radar data. Both types of data have significant strengths and weaknesses. Rain gauge data require less data treatment compared to radar data, and measurements are often available for longer time periods. Rain gauges measure rainfall at ground level, which is the rainfall of interest in hydrological modelling (Thorndahl et al., 2016), and often have a temporal resolution of around 1 minute (Einfalt et al., 2004).

A major weakness about rain gauge data is the lack of information on rainfall movement (mainly for convective events), spatial variation and coverage. Radar data, on the other hand, gives information about rainfall movement and spatial coverage (Thorndahl et al., 2016), and have significantly improved our understanding of how precipitation is formed (Collier, 1989). Weaknesses of radar data is that rainfall intensities are inferred based on reflectivity with often very high uncertainties for high rainfall intensities. Furthermore, radar data is based on an instantaneous scan of volume high above ground that is then used

to represent the average rainfall intensity during the entire sampling time. This can lead to aggregation errors and might not reflect the rainfall at ground level (Einfalt et al., 2004).

Weather Generators (WG) to simulate rainfall are numerous and diverse in kind, input data, spatial and temporal scale (Arnbjerg-Nielsen et al., 2013; Wilks and Wilby, 1999). In the field of rainfall simulation, focus has until now been on models

based on rain gauge data and hence several WGs model precipitation as a stochastic point process (Burton et al., 2008; Cowpertwait and O'Connell, 1997; Onof and Arnbjerg-Nielsen, 2009). Weather generators can be based upon a dense network of rain gauges in order to include some spatial variation in the model (e.g. Jinno et al. (1993); Willems (2012); Sørup et al. (2016)) but none of these WGs describe the spatial dynamics of rainfall at a resolution suitable for urban hydrology.

Radar data has been suggested as potential rainfall input in urban hydrology since the mid 80's (Einfalt et al., 2004). Radar products have more recently become available in spatial and temporal resolution fulfilling the resolution requirement in urban hydrology, and has within the last 1-2 decades become more frequently used in urban hydrology along with increasing length of recording period (Thorndahl et al., 2016). Weather generators based on high resolution radar data are very limited, and often rely heavily on statistically based variables (e.g. Peleg and Morin, (2012)). The downside of this is the absence of physically

based variables to represent the spatio-temporal variation in rainfall and thereby enabling linking WGs to e.g. climate change models.

This study aims to quantify and describe spatial rainfall as a function of temporal and spatial dynamics, rainfall types and seasonal variation. Descriptive statistical methods are applied to analyse selected physically based variables and their internal

correlation. The study aims to statistically describe spatio-temporal varying rainfall using physically based variables, in order to assess the possibility of creating artificial spatio-temporal rainfall series, of high resolution scales useful for urban hydrology.



## 2 Data and case area

### 2.1 Case area

The case area is a 38x48km rectangle (1824 km$^2$) surrounding the catchment of the river Wupper in North Rhine-Westphalia, Germany. It stretches from the Rhine lowland in southwest to the more hilly area in east, with steep valleys around the river

Wupper. The elevation varies from 31 meters to 483 meters above sea level (see Figure 1). The mean annual precipitation in the area ranges from 770 mm to 1352 mm, due to strong orographic effects, with lowest precipitation in low lying areas and most precipitation in the highest elevated areas (orographic rainfall). Due to partly high urbanisation, small-scale but highly intense convective rainfall causes flash floods with a huge damage potential, Therefore, a good knowledge about the structures and impacts of different storm types on a high resolution is essential for planning and forecasting matters.

**2.2 Data**

Radar data from the Deutsche Wetterdienst (DWD) Doppler C-band radar network was used in this study (5-minute temporal resolution, 1x1 km spatial resolution). The data comes from the Wupper Association and spans 13 years, from the 1[st] of November 2000 to the 1[st] of November 2013. The case area is within the range of the Essen radar and partly within the range of the Flechtdorf and Neuheilenbach radars. The data is a weighted composition of the three radars (Einfalt and Lobbrecht,

2011).

The data is post processed by hydro & meteo GmbH & Co. KG on behalf of the Wupper Association. Data is corrected in regards of blockage, clutter and attenuation. The reflectivity (Z) rainfall intensity (R) relationship is fixed as $Z = 256 \cdot R^{1.42}$. In the Wupper Association district the radar data is adjusted to rain gauge data on a daily basis, with a correction factor per

20 gauge in a 1 km correction grid using inverse distance weighting. Rain gauge data is beforehand visually inspected and compared to nearby gauges in order to secure the quality. There are 60 rain gauges within the area of the Wupper Association. The post processed data have less than 5 % difference from annual ground truth (Frerk et al., 2012)

## 3 Methodology

### 3.1 Extreme events

Extreme events are identified based on time series data and defined based on a Peak Over Threshold method (Coles, 2001). A Type II censoring is applied with a prefixed number of 39 extreme events, equal to an average of 3 events per year (Mikkelsen et al., 1995). Two types of extreme events are considered, 1-hour and 24-hour extreme events, based on the maximum average intensity for either 1 hour or 24 hours.



## 3.2 Spatial selection of extreme events

The extreme event definition is based on time series of point data. To our knowledge there is no generally applied procedure to sample extreme events from multisite or areal measurements such as radar data. Based on time series data we examine four methods to identify rain events in order to determine the number of grid cells which should be considered when selected

extreme events for further analyses. All methods identify the number of rain events in the data period, average length of rain events, average maximum number of grid cells registering each event, and seasonal distribution of rain events. The sampling strategies (SS) are listed below in order of increasing number of grid cells:

SS1.    Sampling from 1 grid cell

SS2.    Sampling from 5 grid cells

SS3.    Sampling from one side of the mountains (every ninth grid cell)

SS4.    Sampling from the entire catchment (every ninth cell)

*SS1, Sampling from 1 grid cell*

The simplest sampling strategy is choosing one grid cell from which rain events are identified. Rain events separated by dry

periods less than 24 hours apart are aggregated to one event in accordance with (Madsen et al., 2002, 2009). The single grid cell considered is shown in Figure 2.

*SS2, Sampling from 5 grid cells*

The second sampling strategy considers 5 grid cells in a spatially small area on the same side on the mountain as the

20 predominant wind direction (west wind, see Figure 2). Precipitation occurs when at least one of the locations measures rainfall and events are aggregated using the same approach as when sampling from one grid cell.

*SS3, Sampling from one side of the mountain (every ninth grid cell)*

The third sampling strategy for rain events considers a larger part of the catchment. Due to the strong orographic effect, the

25 part of the catchment which is on the same side of the mountains and therefore located on the same side where the main part of the weather arrives from is considered. Every ninth grid cell in this area is selected with a total of 104 grid cells (see Figure 2). Precipitation events are defined using the same approach as when sampling from 5 grid cells.

*SS4, Sampling from the entire catchment (every ninth grid cell)*

The fourth sampling strategy concerns the entire catchment. Every ninth grid cell is selected, and precipitation time series are merged with a 24-hour dry period between independent events. A total of 187 grid cells are considered.





### 3.3 Data analysis

### 3.3.1 Seasonal variation

The seasonal variation of occurrence of extreme events from the five grid cells filled in Figure 2 is analysed for 1-hour and 24-hour extreme events. The analysis is based on four seasons: winter (December-February), spring (March-May), summer

(June-August) and fall (September-November).

### 3.3.2 Spatial correlation

The spatial correlation between 4950 pairs of grid cells (100 randomly selected grid cells) is calculated, applying the framework of spatial correlating structures by Mikkelsen et al. (1996). The method calculates the spatial correlation by estimating the correlation of extreme events that are meteorologically dependent. The unconditional correlation coefficient ρ between a pair

of grid cells (*A* and *B*) is calculated by identifying concurrent events. If it is assumed that the start ($t_s$) and end ($t_e$) times of all events are known, concurrence between the *i*'th event at grid cell A, $Z_{Ai}$ and the *j*'th event at grid cell B, $Z_{Bj}$ is defined as:

$$\{Z_{Ai}, Z_{Bi}\}: \left[t_{si} - \tfrac{1}{2}\Delta t, t_{ei} + \tfrac{1}{2}\Delta t\right]_A \cap \left[t_{si} - \tfrac{1}{2}\Delta t, t_{ei} + \tfrac{1}{2}\Delta t\right]_B \neq \varnothing \qquad (1)$$

where $\Delta t$ is a lag time introduced to ensure that events can be concurrent events though travelling time means that these events do not overlap in time. $\Delta t$ was in this study set to 11 hours equal to the $\Delta t$ used in Gregersen et al. (2013). Based on the sample

of concurrent events and the sample of not concurrent events in a pair of grid cells, the unconditional covariance is estimated as:

$$Cov\{Z_A, Z_B\} = Cov\{E\{Z_A|U\}, E\{Z_B|U\}\} + E\{Cov\{Z_A, Z_B|U\}\} \qquad (2)$$

Given by the definition in Mikkelsen et al. (1996), the unconditional correlation coefficient $\rho$ can now be estimated by dividing the unconditional covariance with the standard deviation for the two stations. Following the procedure proposed by Gregersen

et al. (2013) the data is hereafter divided into bins based on distance between stations and the average $\rho$ for each bin is calculated in order to minimise noise in the data set. An exponential function is fitted to data, relating the distance between a pair of stations with the unconditional correlation coefficient $\rho$. The e-folding distance is then found as the distance where the unconditional correlation have decreased to 1/e, based on the fitted exponential function (Gregersen et al., 2013).

### 3.3.3 Spatial variation

The spatial variation in extreme events is analysed for the five grid cells filled in Figure 2. The extreme events sampled from each of the five grid cells, are compared to identify the small-scale variability in sampled extreme events. The black filled grid cell is used as reference and compared to the four grey filled grid cells. The number of concurrent events and the distance between the grid cells are calculated.



### 3.4 Characterisation of events

The two sets of 39 extreme events are characterised by 16 variables chosen to describe a variety of event properties (see Table 1); these can be further aggregated into six categories: *Duration*, *intensity*, *wet area coverage*, *depth*, *rain cell properties* and *movement*. Rain cell properties and movement are described with a simple rain cell identification and tracking algorithm as described below.

### 3.4.1 Rain cell identification

Rain cells are identified in each time step by assigning an intensity threshold and an areal threshold. The intensity threshold is set to 25% of the maximum 5-minute intensity for the given event with a minimum threshold of 7 mm h$^{-1}$. The areal threshold is set to a minimum coverage of 10 km$^2$. An event specific threshold is chosen to distinguish between different rain cell types (e.g. convective and front cells) and secure a high threshold for all events which result in a more stable tracking of a clear cell centre (Dixon and Wiener, 1993). Rain cells with an area below the areal threshold are disregarded to avoid noise in the overall tracking from multiple small cells (Dixon and Wiener, 1993). An ellipse is fitted to each of the identified rain cells, with the coordinates for the centroid, length of the axis and orientation in degrees between major axis and east-axis (Belachsen et al., 2017; Peleg and Morin, 2012).

### 3.4.2 Rain cell tracking

Various complex rain cell tracking algorithms can be found in literature (e.g. Dixon and Wiener, 1993; Handwerker, 2002; Kyznarová and Novák, 2009). For describing the overall moving direction and velocity of each rain event this study has developed a simple tracking algorithm. Rain cell movement is recorded by linking the identified rain cells in each time step together in a simple tracking algorithm based on the position of the centroid. Tracking is based on the moving direction and velocity from last time step which is used to predict the approximate position of the rain cell in the next time step. The rain cell with the centroid closest to the predicted position of the rain cells centroid is linked to the rain cell in the previous time step with no further evaluation of the fit. A maximum distance of 7.5 km, corresponding to a moving velocity of 25 m s$^{-1}$, from the predicted position of the rain cell to the linked rain cell is applied. For new rain cells, the position of the rain cell in time step one is the predicted position of the rain cell in next time step. The tracking algorithm manages birth, tracking and death of rain cells. If splitting of a rain cell occurs, the algorithm will treat it as continuous tracking of the rain cell and a birth of a new rain cell. In case of merging of two rain cells, the algorithm will classify it a death of one rain cell and continue tracking of the other rain cell.

### 3.5 Statistical analyses

All statistical analyses are performed using normalised data, i.e. mean zero and variance one. All analyses are carried out in R using the build-in R Stats Package version 3.4.1.





### 3.5.1 Principal component analysis

Principal Component Analysis (PCA) is used to estimate the correlation structures in data and determine the number of dimensions necessary to describe it. PCA is a linear orthogonal transformation method to describe the variance of data using linear combinations, called Principal Components (PC).

Eigenvalues and corresponding vectors are calculated based on the correlation matrix of the normalised data. The eigenvector with the $i^{th}$ largest eigenvalue ($\lambda_i$) is noted the $i^{th}$ principal axis, where $PC_i$ represent the projection of data on the $i^{th}$ principal axis (Morrison, 1967). The percentage of the variance, which $PC_i$ describes, is calculated as the percentage of the sum of the eigenvalues based on the $i^{th}$ eigenvalue (Morrison, 1967).

Two tests are applied to determine the number of dimensions necessary to describe data. The first test is an approximate test to estimate the number of significant PC's based on the magnitude of the eigenvalues. The hypothesis tested is that the last $k+1$ to $m$ eigenvalues are similar and therefore non-significant, where $m$ is the total number of eigenvalues. The test is described in Lawley and Maxwell (1963) and Anderson (1984) as:

15 $\quad H_0: \lambda_1 \geq \cdots \geq \lambda_k \geq \lambda_{k+1} = \cdots = \lambda_m$ (3)

The test statistic is defined as:

$$z_2 = -n * \ln\left(\frac{\prod_{i=k+1}^{m} \lambda_i}{\hat{\lambda}^{m-k}}\right)$$ (4)

where $\hat{\lambda}$ is defined as:

$$\hat{\lambda} = \sum_{i=k+1}^{m} \lambda/(m-k)$$ (5)

20 The second test estimates the number of effective spatial degrees of freedom based on the eigenvalues and was proposed by Bretherton et al., (1999) as:

$$N_{eff} = \frac{\left(\sum_{i=1}^{m} \lambda_i\right)^2}{\sum_{i=1}^{m} \lambda_i^2}$$ (6)

### 3.5.2 Cluster analysis

Partitioning and hierarchical clustering is performed on the dataset to identify similarities between the events based on all 25 variables. Both clustering methods are based on the normalised data, i.e. the same dataset used for the PCA.

The K-means clustering algorithm presented by Hartigan (1975) and Hartigan and Wong (1979) is selected as partitioning clustering method. The method divides the dataset into a predefined number of clusters by minimising the sum of squared distances (Hartigan, 1975). Initially all events are assigned a cluster and it is afterwards for each event tested if the Euclidean 30 distance to the centre of the cluster will be reduced if the event is moved to another cluster. The centre of the cluster is defined as the mean of each of the PCs that the events in the cluster are projected onto and updated every time a cluster is moved from



or added to the cluster. If *l(i)* describes the cluster where the event *i* is contained and *l* represents any cluster then *D[i,l(i)]* denotes the Euclidean distance between event *i* and cluster centre *l(i)* and similarly *D[i,l]* denotes the Euclidean distance between event *i* and the centre of any other cluster. Reallocation of events to another cluster is done if it decreases the error as:

$$\frac{N_l D[i,l]^2}{N_l+1} - \frac{N_l D[i,l(i)]^2}{N_{l(i)}+1} < 0 \tag{7}$$

The hierarchical clustering methods selected are the Ward method and the Average Linkage Method (Murtagh, 1983). The methods are agglomerative clustering methods where all events start in separate clusters, the two less dissimilar events are joined until all events are in one cluster (Cormack, 1971). The dissimilarities between events are calculated based on the Lance-Williams cluster update method (Lance and Williams, 1966). As an example, assume that the dataset is divided into three clusters, *h, i* and *j* with $n_h$, $n_i$, and $n_j$ number of events in each of the clusters and the dissimilarities between the clusters are denoted $d_{hi}$, $d_{hj}$ and $d_{ij}$. The dissimilarities between two events are calculated as the Euclidean distance between the events when all events are in separate clusters. If the dissimilarity between the clusters *i* and *j* is smallest, the two clusters are joined to the new cluster k, with $n_k = n_i + n_j$ events. The dissimilarity between cluster k and h is calculated as:

$$d_{hk} = \alpha_i d_{hi} + \alpha_j d_{hj} + \beta d_{ij} + \gamma(d_{hi} + d_{hj}) \tag{8}$$

Where $\alpha_i$, $\alpha_j$, $\beta$ and $\gamma$ are parameters determined by the clustering method (Cormack, 1971). For the Ward method $\alpha_i = (n_i + n_k)/(n_i + n_j + n_k)$, $\beta = -n_k/(n_i + n_j + n_k)$ and $\gamma = 0$ while for the Average Linkage $\alpha_i = n_i/(n_i + n_j)$, $\beta = 0$ and $\gamma = 0$. The results are illustrated using dendrograms.

### 3.5.3 Stepwise regression

Stepwise regression is used to estimate the number of independent variables which are necessary to describe a dataset (Draper and Smith, 1998). The method fits a linear model which is as simple as possible and as complex as necessary to data. Stepwise regression is performed with a stepwise forward selection of variables to include in the model and an evaluation of all variables in the model to test if any of the included variables should be eliminated (Rawlings et al., 1998). The Akaike's Information Criterion (AIC) (Akaike, 1971) is used to determine the trade-off between simplicity and fit:

$$AIC = -2 \cdot LogLikelihood + 2p \tag{9}$$

Where *p* is the number of parameters in the model, as the lower *AIC* the better fit p therefore acts like a penalty when adding more parameters. An optimal linear model is fitted to each of the 16 variables from the event analysis independently with the remaining variables as descriptors using stepwise regression.



## 4 Results and discussion

### 4.1 Spatial selection of extreme events

For the four sampling strategies, the number of rain events in the data period, average length of rain events, average maximum number of grid cells registering each event, and seasonal distribution of rain events are shown in Table 2. The total number of events within the data period decreases substantially, and the average length increases similarly, when the number of grid cells considered is increased. The seasonal proportion of the events indicates that especially summer events are joined when more grid cells are considered. The results from Table 2 indicate that meteorologically independent events are joined when grid cells in a large part of the catchment is considered. In order to sample events which are meteorologically independent and use the knowledge about extreme events from rain gauge data, sampling method one, only considering one grid cell when sampling extreme events, is selected.

### 4.2 Data analysis

#### 4.2.1 Seasonal variation

The seasonal variation of occurrence of extreme events for each of the five grid cells filled in Figure 2, can be seen in Figure 3. There is very little variation between the five stations in seasonal variation in occurrence of extreme events. In contrast to this, the difference between 1-hour and 24-hour extreme events is more pronounced, with 1-hour extreme events almost only occurring in the summer while 24-hour extreme events are more uniformly distributed over the year. This corresponds well with the seasonal difference in precipitation in the area (ExUS, 2010; Quirmbach et al., 2012) and the expectance of differences in seasonal variation between different event types, convective vs. front events, (Gregersen et al., 2013).

#### 4.2.2 Spatial correlation

The spatial correlations calculated between 4590 pairs of grid cells for 1-hour and 24-hour extreme events are shown in Figure 4. The spatial correlation for 1-hour extreme events decreases faster with distance than for the 24-hour extreme events. This indicates that 1-hour extreme events are more localised small-structured events while the 24-hour extreme events are spatially larger events. From the fitted exponential functions, the e-folding distances are calculated to be 9.3 km and 21.3 for 1-hour and 24-hour extreme events respectively. Studies calculating e-folding distances on rain gauge data show similar orders of magnitude and differences between 1-hour and 24-hour extreme events (Gregersen et al., 2013).

#### 4.2.3 Spatial variation

The spatial variation of extreme events is indicated by calculating the similarity of choice of events by grid cells close to the chosen grid cell, see Table 3. Only approximately 55 % of the 1-hour extreme events are the same events for the four surrounding grid cells (grey filled, Figure 2), when comparing to the reference grid cell (black filled, Figure 2), while 80% of



the 24-hour extreme events are the same. This again indicates that 1-hour extreme events are very localised events and state the importance of carefully selecting a sampling strategy for analysed extreme events.

## 4.3 Event characterisation

1-hour and 24-hour extreme events sampled using sampling method one and described by the chosen 16 variables in the event analysis can be found in the supplementary material. The 39 sampled 1-hour extreme events consist of 27 summer events and 12 non-summer events while the 39 24-hour extreme events consist of 9 summer events and 30 non-summer events. Differences between variables describing 1-hour and 24-hour extreme events are in particular pronounced for the variables *Duration*, *Maximum 10 minute intensity* and *Maximum depth*, which can be related to the differences between convective events and events within front systems. The results from the event characterisation, in relation with the results from the seasonal variation and spatial correlation indicate that the events sampled are representative for extreme events over the year in the area. Twelve events occur in both 1-hour and 24-hour extreme events; these are listed in Table 4.

## 4.4 Statistical analyses

### 4.4.1 Principal component analysis

The PCA is performed on both the 1-hour and 24-hour extreme event dataset separately and as one dataset. The shown results focus on the 1-hour and 24-hour extreme events treated as one dataset. In Table 5 the weighted composition of variables in each of the first nine PCs can be seen. $PC_1$ and $PC_2$ are influenced by most of the variables describing the means of the events. $PC_3$ describe the movement of the rain cells in the events and is mostly influenced by *Standard deviation of direction* and *Mean direction*, which is not contributing much in $PC_1$ and $PC_2$, and *Standard deviation of velocity*. $PC_4$ can be summarised to describe the extremity of the events, mostly influenced by *Duration, Ratio 10min, Maximum 24hr intensity, Mean wet area* and *Rain cell life time*. The two first PCs explain 57.58 % of the total variance and the first nine PCs should be considered if 95 % of the variance must be explained. Based on the eigenvalues 14 PCs are significant when using the approximate test in Eq. (3-5). The alternative test suggests that there are 5.1 effective PCs. As such, five to nine dimensions should be, and up to 14 dimensions could be, considered in order to describe the variability of the events when considering both 1-hour and 24-hour extreme events.

In Figure 5 the dataset is projected into the first two PCs. In the left figure the PCA is performed on the 24-hour dataset alone and the 1-hour extreme events are projected on to the two first PCs for the 24-hour extreme events, i.e. the 1-hour and 24-hour dataset are normalised separately. On the right the PCA is performed on the combined dataset, i.e. the 1-hour and 24-hour dataset are normalised together. While no distinct difference between 1-hour and 24-hour extreme events can be seen when data is normalised separately (Figure 5, left), a clear clustering between 1-hour and 24-hour extreme events can be seen when data is normalised as one dataset and a combined PCA is performed (Figure 5, right). This indicates that the observed



differences between 1-hour and 24-hour extreme events can be described by scaling across the variables rather than a change in the overall structure of the spatial precipitation. Furthermore, there is a tendency of increasing extremity with decreasing PC1 and PC2. How able the combined dataset is to distinguish between 1-hour and 24-hour extreme events for PC 1-7 is illustrated in Figure 6. The distinctions between the two event types are clear for all combination of PCs if either PC1 or PC2

is included.

The seasonal variation of the sampled extreme events visualised by the two first PCs can be seen in Figure 7. A distinction between summer and non-summer events is clear, reflecting the difference in the seasonal variation between 1-hour and 24-hour extreme events.

**4.4.2 Cluster analysis**

The K-means clustering algorithm is performed with a predefined number of two and four clusters (Figure 8). The two clusters in Figure 8 left, describe the distinction between 1-hour and 24-hour extreme events with few 1-hour extreme events in the first cluster primarily consisting of 24-hour extreme events and opposite in the second cluster. In Figure 8 right, the two very extreme events on the 06-08-2009 and the 19-06-2013 constitute a separate cluster. These events can be described as severe

convective events with very high *Maximum 10 minute* and *Maximum 1 hour intensity* and high *Ratio between maximum and average 10 minute intensity*. The second cluster contains 1-hour extreme events or events sampled as both types of extreme events. The extreme events in the second cluster are clearly convective events, less extreme than in the first cluster but with remarkably high *Maximum 1-hour intensity* and a large *Ratio between maximum and mean depth*. The third cluster is dominated by 1-hour extreme events coupled with 24-hour extreme events. Common for all 24-hour extreme events in this cluster is a

shorter *Duration* compared to the rest of the 24-hour extreme events and a relative low *Mean depth*. The events in cluster three can be characterised as convective events that could be within a front system. The last cluster, cluster four, consists mostly of 24-hour events with characteristics as long *Duration*, low *Ratio between maximum and mean depth* and low *Maximum 1 hour intensity*. These events can be classified as frontal events with little or no convective activity. Two 1-hour extreme events are within this cluster; these differ from the rest of the 1-hour extreme events by long *Duration*, and large *Minimum* and *Mean*

*depths*.

The dendrograms of the two hierarchical clustering methods Ward and Average linkage can be seen in Figure 9. Even though the two dendrograms appear quite different, many similarities can be seen between the two methods when a specific number of clusters is chosen. Figure 10 visualises the first four clusters from the dendrograms. The clusters identified with the Ward

method are very similar to the clusters using the K-means clustering method. The clusters identified with the Average linkage method have the same tendency as the other clustering method, though with more focus on the unusual events. If the number of clusters are increased, the Average linkage method will divide the 1-hour and 24-hour extreme events into separate clusters





just like the other clustering method. For the purpose of modelling spatial rainfall the different types of extreme events described by the K-means clustering method seem more appropriate than the Ward clustering method.

### 4.4.3 Stepwise regression

Linear models for all of the variables are identified by means of stepwise regression. Table 6 specifies the variables in each of the linear models and the goodness of the fit while Table 7 summarises the number of times that each variable is used in the linear models. The linear models in Table 6 consist of between 3 and 10 descriptive variables, with an average of 7.4, which correspond well with the number of independent variables found in the PCA. Six variables are used in more than half of the linear models: *Ratio 10 minute*, *Maximum 24 hour intensity*, *Mean wet area*, *Ratio depth*, *Cell lifetime* and *Standard deviation of velocity*. These variables were also identified as variables that are able to distinguish between 1-hour and 24-hour extreme 10 events. Furthermore, do the six variables correspond well with the variables influencing the first four PCs in the PCA: *Ratio 10min* and *Ratio depth* both largely influence PC1, *Maximum 24 hour intensity* influence PC2, *Standard deviation of velocity* influence PC3 and *Mean wet area* and *Cell lifetime* influence PC4.

### 5 Conclusion

Spatial rainfall from radar data was analysed using a range of metrics and the findings are in general in accordance with current 15 understanding of spatial rainfall. The seasonal variation and spatial correlation of the analysed extreme events confirm a clear difference between 1-hour and 24-hour extreme events which can be described as a difference between convective and front events. Four sampling strategies for sampling spatial extreme events were analysed and it was found that it is best to sample from only one grid cell to avoid unrealistic long events with sub-events that are meteorologically independent. 1-hour extreme events are very local. It was shown that a least 50 % of sampled extreme events would change if another grid cell within a 20 radius of approximately 5 km were chosen as sample point, with decreasing similarity in sampled extreme events with increasing distance between the compared grid cells. This study suggests that further development on a sampling strategy for sampling spatial extreme events is needed.

Events were characterised by 16 variables giving a thorough description of the spatio-temporal variability of the events. All 25 variables contribute with information about the analysed extreme events, even though there are correlations suggesting that not all dimensions are necessary. The PCA suggest five to nine dimensions necessary to describe the data, but up to 14 PC's were found significant, implying that more dimensions might be relevant to consider. From the PCA and cluster analysis it was possible to distinguish between 1-hour and 24-hour extreme events and identify four different storm types with varying level of extremity. Variables found important in the PCA and cluster analysis were concluded to be important in the stepwise 30 regression as well. The findings of the three methods supplement each other well and no contradictions between them have been identified.



A simple rain cell identification and tracking algorithm was developed for the study to describe the overall tendency in rain cell lifetime, number, direction and velocity of extreme events. For the purpose of this study the relatively simple algorithm proved to be sufficient to give a realistic picture of the related variables.

The results of this study conduct the preliminary steps prior to setting up a weather generator with similar properties as high-resolution radar rainfall data. The results point out which variables in such a weather generator should be considered independent and which could be co-varied. The study should be considered as a first step into the direction of good practises to find and analyse single storm events in radar rainfall data sets. The tested methods helps to understand the characteristics of

the different storm types and the results show which weather regimes should be included to reproduce all relevant situations in which extremes could appear.

**Acknowledgement**

The radar data used is a quality-controlled composite product from hydro & meteo GmbH&Co.KG based on the polar data product from the Deutscher Wetterdienst (DWD) radar network. Data is owned by the Wupperverband and has been made

freely available for this research purpose. Inquiries regarding the data should be addressed to the Wupperverband.



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





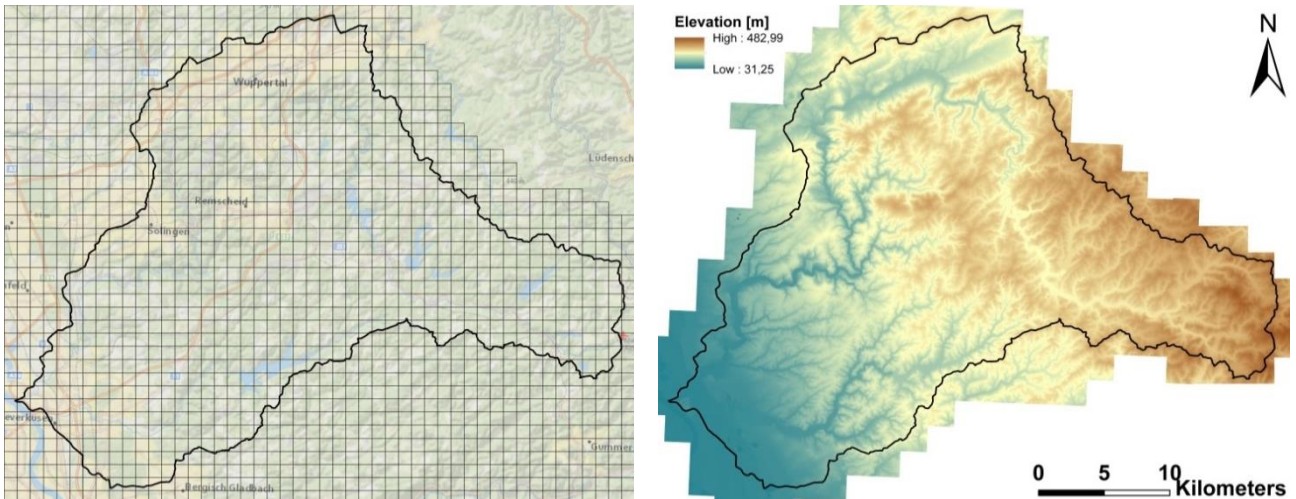

**Figure 1: Overview of the case area. Left: Gridded area represents the part of the catchment where time series data is produced. Right: Elevation in the Wupper catchment.**

**Figure 2: Overview of the four sampling strategies: - SS1: Filled black cell, SS2: Black and grey filled cells, SS3: cells outlined in light grey and SS4: cells outlined in light and dark grey.**



**Table 1: Description of variables**

| Variable | Short name | Unit | Description |
|---|---|---|---|
| Duration | Duration | Hours | From start to end with an extension of 2 hours in each end to consider the event in the entire case area. |
| Intensity | Max 10min | mm min$^{-1}$ | Maximum average intensity for 10 minutes. |
|  | Ratio 10min | - | Ratio between max 10 minute and mean 10 minute intensity. |
|  | Max 1h | mm min$^{-1}$ | Maximum average intensity for 1 hour. |
|  | Max 24h | mm min$^{-1}$ | Maximum average intensity for 24 hours. |
| Wet Area | Mean wet A | - | Average ratio of cells with precipitation (wet cells) from each time step of the event. |
| Depth | Min depth | mm | Value of the grid cell with the lowest depth in the case area. |
|  | Max depth | mm | Value of the grid cell with the highest depth in the case area. |
|  | Mean depth | mm | Average depth considering all cells in the case area. |
|  | Ratio depth | - | Ratio between max depth and mean depth. |
| Rain cell properties | Cell num | - | Number of tracked rain cell in the rain event. |
|  | Cell life | min | Average lifetime of the rain cells in the event. |
| Movement | Mean vel | m s$^{-1}$ | Mean rain cell velocity |
|  | Sd vel | m s$^{-1}$ | Standard deviation of velocity. |
|  | Mean dir | Degrees | Mean moving direction of rain cells, compass degrees. |
|  | Sd dir | Degrees | Standard deviation of direction. |

**Table 2: Results from the four sampling strategies described in Sect. 3.2**

| Sampling strategy | Number of events total | Average event length [h] | Average number of grid cells | Proportion of events in winter | Proportion of events in spring | Proportion of events in summer | Proportion of events in fall |
|---|---|---|---|---|---|---|---|
| SS1 (1 grid cell) | 879 | 65.29 | 1.00 | 0.21 | 0.27 | 0.28 | 0.24 |
| SS2 (5 grid cells) | 843 | 80.61 | 4.38 | 0.22 | 0.27 | 0.27 | 0.25 |
| SS3 (west side of the mountain) | 439 | 219.88 | 67.74 | 0.28 | 0.27 | 0.18 | 0.27 |
| SS4 (total catchment) | 297 | 347.48 | 122.61 | 0.31 | 0.27 | 0.16 | 0.26 |





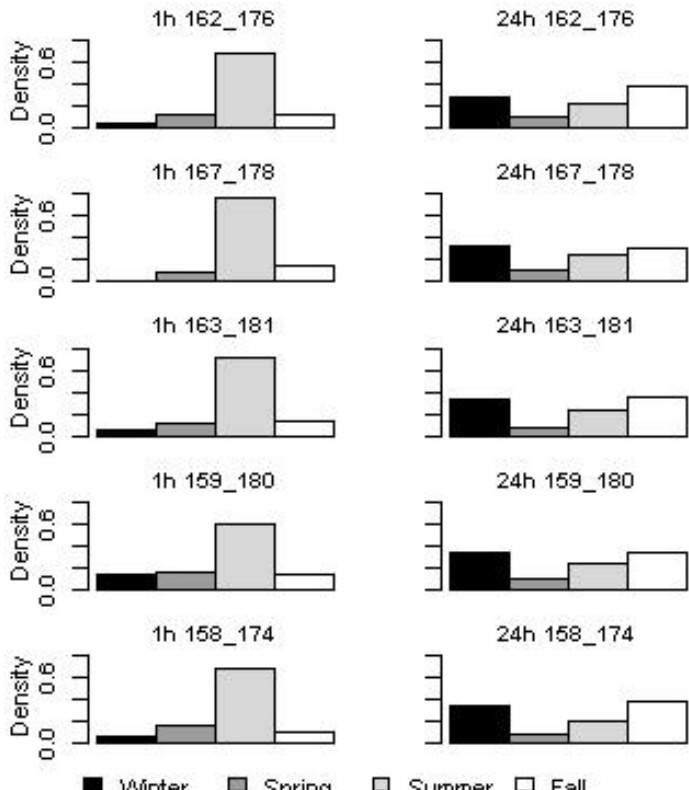

**Figure 3: Seasonal variation in occurrence of extreme events for each of the five grid cells filled in Figure 2.**





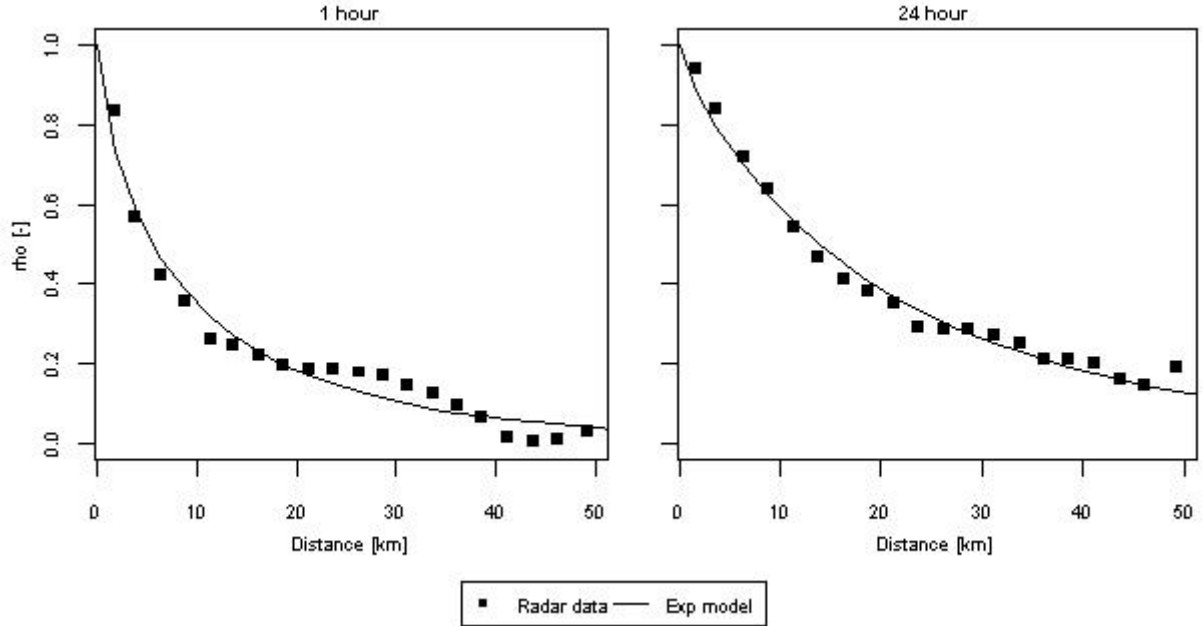

**Figure 4: Spatial correlation calculated for binned data of 100 grid cells, 4950 pairs.**

**Table 3: Comparison of the five grid cells filled in Figure 2, the black filled grid cell is used as reference.**

| Name | 162_176 | 167_178 | 163_181 | 159_180 | 158_174 |
|---|---|---|---|---|---|
| distance (S,E) [km] | (0,0) | (-4,-1) | (-3,7) | (3,5) | (4,-3) |
| distance [km] | 0.00 | 5.39 | 5.10 | 5.00 | 4.47 |
| Similar 1-hour | 39 | 19 | 21 | 23 | 22 |
| Similar 24-hour | 39 | 31 | 32 | 30 | 32 |
| Similar 1-hour [%] | 100 | 49 | 54 | 59 | 56 |
| Similar 24-hour [%] | 100 | 79 | 82 | 77 | 82 |





**Table 4: Overview of the 12 events sampled as both 1-hour and 24-hour extreme event. The numbers refer to the tables in the supplementary material.**

| Date (LT) | 1-hour | 24-hour |
|-----------|--------|---------|
| 17-07-2001 | 3 | 2 |
| 19-08-2002 | 6 | 6 |
| 05-10-2002 | 7 | 7 |
| 10-09-2004 | 12 | 14 |
| 29-06-2005 | 16 | 19 |
| 29-09-2005 | 19 | 22 |
| 06-08-2007 | 22 | 25 |
| 18-08-2007 | 23 | 26 |
| 10-08-2010 | 29 | 30 |
| 19-06-2013 | 36 | 37 |
| 22-07-2013 | 37 | 38 |
| 06-09-2013 | 39 | 39 |

5    **Table 5: Composition of variables for the first nine PC's in a combined PCA including both 1-hour and 24-hour extreme events.**

|  | PC1 | PC2 | PC3 | PC4 | PC5 | PC6 | PC7 | PC8 | PC9 |
|--|-----|-----|-----|-----|-----|-----|-----|-----|-----|
| Duration | 0.25 | -0.27 | 0.10 | -0.40 | 0.21 | -0.11 | 0.09 | -0.30 | 0.06 |
| Max 10min intensity | -0.29 | -0.31 | 0.02 | -0.17 | -0.23 | 0.01 | -0.18 | 0.16 | 0.16 |
| Ratio 10min | -0.28 | -0.27 | 0.05 | -0.33 | -0.23 | -0.06 | -0.10 | -0.10 | 0.10 |
| Max 1h intensity | -0.32 | -0.31 | -0.10 | -0.02 | -0.10 | 0.17 | -0.07 | -0.06 | 0.17 |
| Max 24h intensity | -0.10 | -0.40 | -0.16 | 0.27 | -0.08 | 0.15 | -0.31 | 0.32 | -0.27 |
| Mean wet area | 0.20 | -0.04 | 0.11 | 0.62 | -0.27 | -0.04 | -0.17 | -0.19 | -0.15 |
| Min depth | 0.32 | -0.30 | 0.10 | 0.01 | 0.15 | -0.13 | 0.03 | -0.08 | 0.01 |
| Max depth | 0.16 | -0.46 | 0.05 | 0.04 | 0.11 | 0.02 | 0.01 | -0.07 | -0.18 |
| Mean depth | 0.30 | -0.35 | 0.08 | 0.00 | 0.11 | -0.09 | 0.01 | -0.03 | -0.12 |
| Ratio depth | -0.33 | 0.01 | -0.20 | -0.05 | 0.07 | 0.25 | 0.23 | -0.46 | -0.57 |
| Number of rain cells | 0.26 | -0.03 | -0.22 | -0.12 | 0.25 | 0.63 | 0.14 | 0.46 | 0.02 |
| Rain cell life time | -0.19 | -0.21 | 0.23 | 0.41 | 0.06 | 0.25 | 0.54 | -0.14 | 0.50 |
| Mean velocity | 0.30 | 0.08 | 0.20 | -0.16 | -0.46 | 0.19 | -0.17 | -0.03 | 0.18 |
| Sd velocity | 0.28 | 0.03 | -0.33 | -0.06 | -0.30 | 0.43 | -0.12 | -0.46 | 0.13 |
| Mean direction | 0.15 | -0.11 | -0.37 | -0.06 | -0.53 | -0.29 | 0.61 | 0.23 | -0.14 |
| Sd direction | 0.02 | -0.05 | -0.71 | 0.16 | 0.25 | -0.29 | -0.18 | -0.13 | 0.38 |
| Proportion of variance | 33.94 | 23.57 | 8.81 | 8.64 | 7.14 | 5.24 | 3.50 | 2.93 | 2.55 |
| Prop. of variance cumulative | 33.94 | 57.51 | 66.32 | 74.97 | 82.11 | 87.35 | 90.85 | 93.78 | 96.32 |




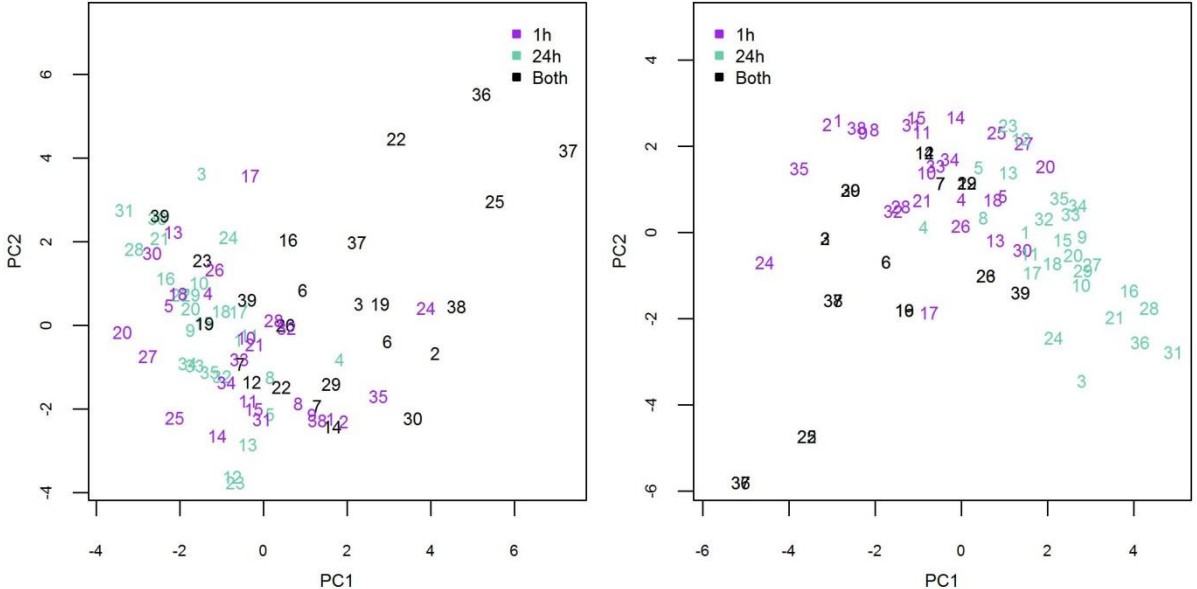

**Figure 5: Projection of the extreme events into the two first PC's. Numbers refer to the tables in the supplementary material. Events sampled both as 1-hour and 24-hour extreme events are marked in black, 1-hour extreme events are marked in purple and 24-hour extreme events are marked in blue. Left: Principal component analysis performed on 24-hour dataset and 1-hour extreme events are projected into the same coordinate system. Right: 1-hour and 24-hour extreme events are treated as one dataset, with a combined PCA.**





**Figure 6: Extreme events projected into PC1 to PC7. 1-hour extreme events are shown in purple, 24-hour extreme events in blue and event sampled with both sampling methods in black.**



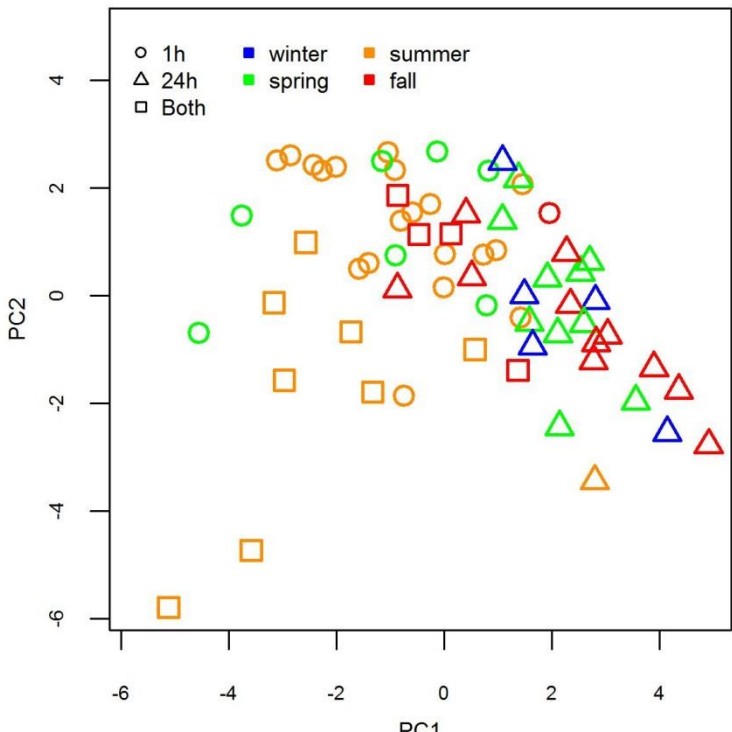

**Figure 7: Projection of extreme events into the two first PC's for the combined dataset. Colours indicate season (winter, spring, summer and fall) and shape indicate extreme event type.**

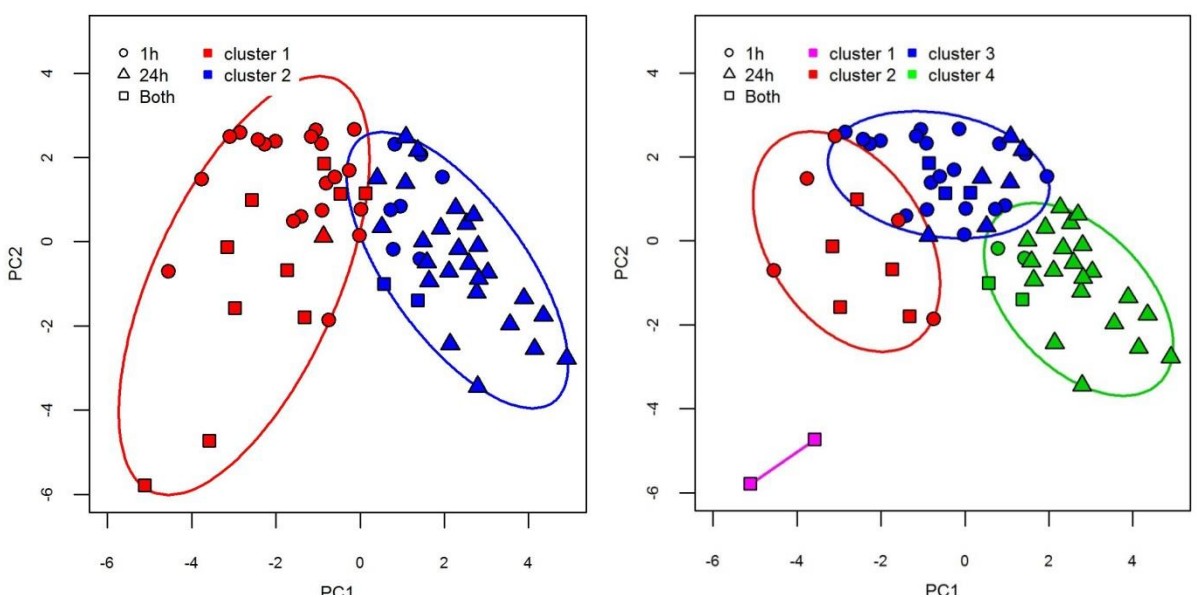

**Figure 8: K-means cluster analysis performed on 1-hour and 24-hour data. Left: Pre-defined number of two clusters. Right: Pre-defined number of four clusters.**





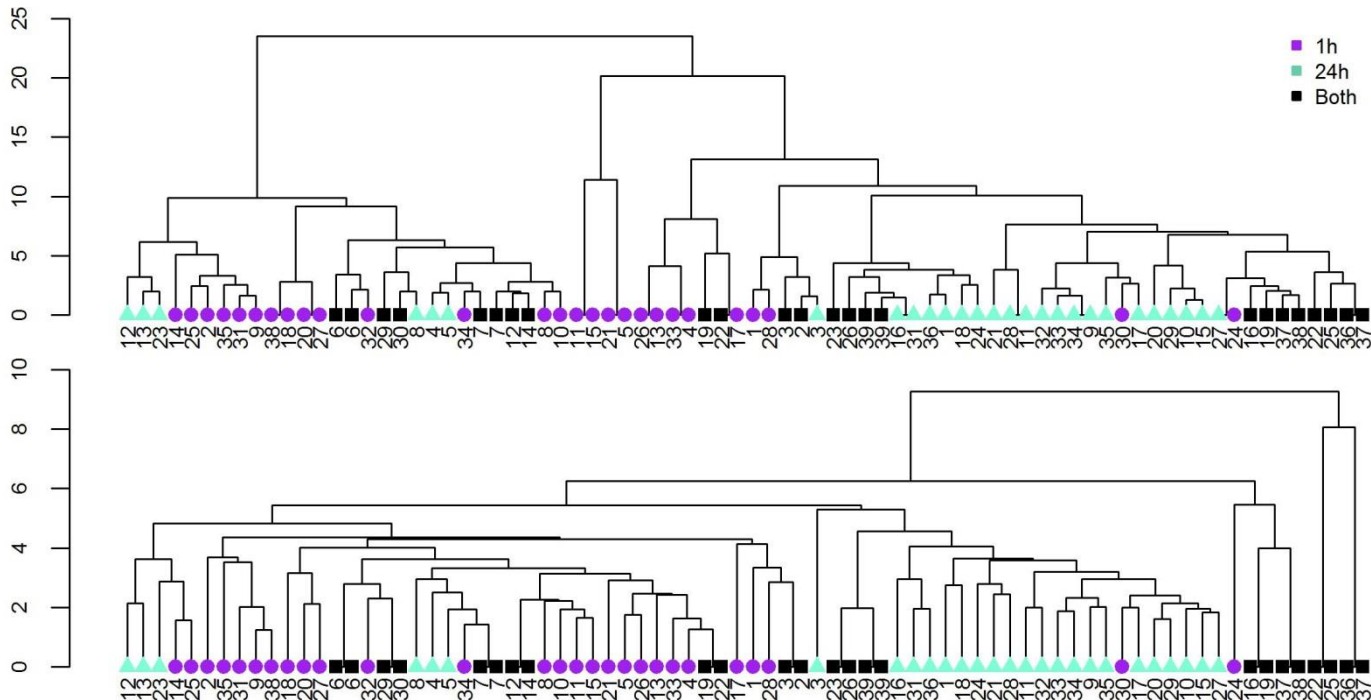

**Figure 9: Dendrogram performed on 1-hour and 24-hour extreme events with Ward (top) and Average linkage (bottom) method.**




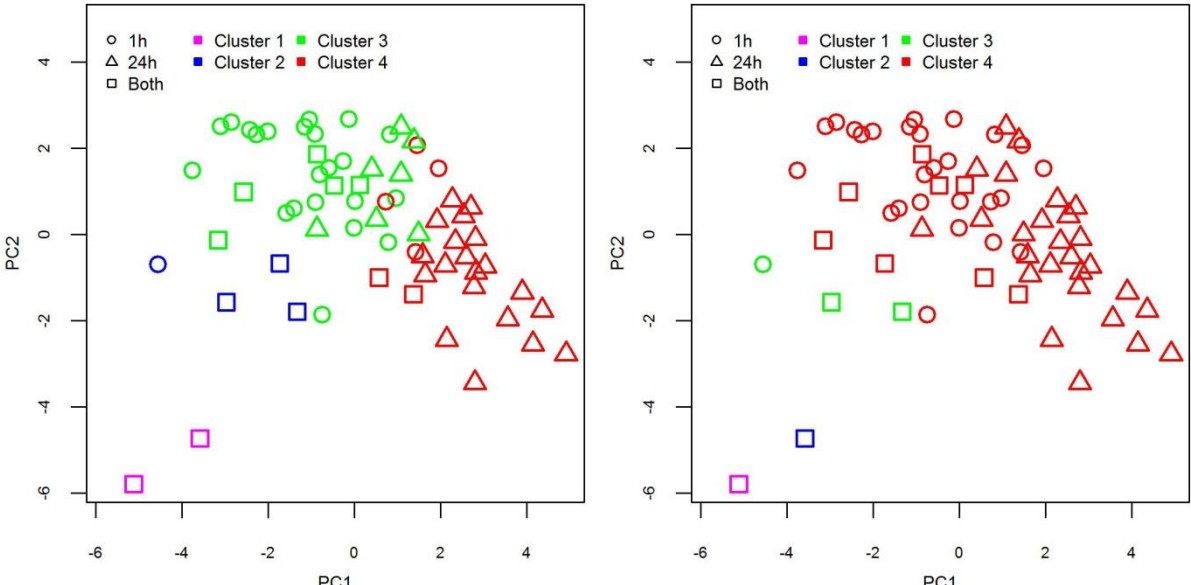

**Figure 10: Graphical representation of the first four clusters from the hierarchical clustering methods. Left: Ward. Right: Average linkage.**



**Table 6: Linear models for each of the variables, using stepwise regression to simplify the models.**

| | Variables, (number of variables) | AIC | $r^2$ |
|---|---|---|---|
| Duration | (8) min depth, mean wet a, mean vel, ratio 10min, max 10min, max depth, max 24h, cell life | -157.46 | 0.88 |
| Max 10min | (10) max 1h, ratio 10min, ratio depth, duration, mean wet a, max 24h, sd dir, cell life, mean depth, cell num | -250.12 | 0.96 |
| Ratio 10min | (9) max 10min, max 24h, duration, ratio depth, mean wet a, mean depth, cell life, sd vel, mean dir | -194.83 | 0.93 |
| Max 1h | (9) max 10min, ratio depth, cell life, sd dir, sd vel, mean dir, max 24h, ratio 10 min, mean vel | -273.07 | 0.97 |
| Max 24h | (6) max 1h, max depth, duration, cell life, ratio 10min, max 10min | -133 | 0.83 |
| Mean wet A | (10) ratio depth, ratio 10min, max 24h, max 10min, duration, max depth, min depth, mean vel, cell num, max 1h | -69.44 | 0.64 |
| Min depth | (6) mean depth, max depth, cell life, duration, mean vel, mean wet A | -229.14 | 0.95 |
| Max depth | (10) mean depth, max 1h, ratio depth, max 24h, cell life, duration, min depth, mean dir, mean wet A, sd vel | -284.67 | 0.98 |
| Mean depth | (9) min depth, max depth, max 1h, cell life, ratio depth, mean dir, max 24h, ratio 10min, max 10min | -323.57 | 0.99 |
| Ratio depth | (7) mean vel, max depth, mean depth, sd vel, max 24h, cell life, sd dir | 83.48 | 0.69 |
| Cell num | (4) sd vel, ratio 10min, mean wet A, max depth | -36.61 | 0.41 |
| Cell life | (10) max 1h, ratio 10min, sd dir, max 10min, ratio depth, sd vel, mean dir, max 24h, mean wet A, min depth | -86.49 | 0.71 |
| Mean vel | (4) ratio depth, sd vel, sd dir, cell life | -84.71 | 0.68 |
| Sd vel | (5) mean vel, sd dir, cell num, ratio depth, mean wet A | -56.58 | 0.55 |
| Mean dir | (6) sd vel, mean depth, sd dir, ratio 10min, max 1h, duration | -19.33 | 0.28 |
| Sd dir | (5) mean vel, sd vel, ratio depth, mean dir, cell life | -28.15 | 0.35 |
| **Average** | | **-133.98** | **0.74** |



**Table 7: Number of times each of the variables is used, when setting up linear models for all variables.**

|  | Times used |
|---|---|
| Duration | 7 |
| Max 10min intensity | 7 |
| Ratio 10min | 9 |
| Max 1h intensity | 7 |
| Max 24h intensity | 9 |
| Mean wet area | 8 |
| Min depth | 5 |
| Max depth | 7 |
| Mean depth | 6 |
| Ratio depth | 10 |
| Number of rain cells | 3 |
| Rain cell life time | 11 |
| Mean velocity | 7 |
| Sd velocity | 9 |
| Mean direction | 6 |
| Sd direction | 7 |