# Peer review of "Explorative Analysis of Long Time Series of Very High Resolution Spatial Rainfall"

_Hydrology and Earth System Sciences, 2018_

## Referee Comment (RC1) · Anonymous Referee #1 · 17 May 2018

The paper "Explorative Analysis of Long Time Series of Very High Resolution Spatial Rainfall" by Emma Dybro Thomassen et al. examines characteristics of extreme rainfall events studied using data from a C-band weather radar composite. The methods used by the authors are not novel, yet I found the analysis and results interesting and I believe they will be of interest for the readers of HESS as well. The paper is well structured and the analysis is sound. However, I found several issues that, although none of them are "major", I would like to bring to the attention of the authors. Please see my comments and suggestions below.

[page lines]

[2 18] "…but none of these WGs describe the spatial dynamics of rainfall at a resolution suitable for urban hydrology" – I disagree! If you review the recent literature you will find at least 3 different rainfall generators that are generating gridded rainfall in a resolution that is finer than kilometer and 5 min. See, for example: Benoit et al. (2018) - Stochastic rainfall modelling at sub-kilometer scale, Peleg et al. (2017) - Partitioning the impacts of spatial and climatological rainfall variability in urban drainage modeling, and Peleg and Morin (2014) - Stochastic convective rain-field simulation using a high-resolution synoptically conditioned weather generator.

[2 24] "The downside of this is the absence of physically based variables to represent the spatio-temporal variation in rainfall and thereby enabling linking WGs to e.g. climate change models" – I think there is also some work done in this direction. See Paschalis et al. (2013) – "A stochastic model for high-resolution space-time precipitation Simulation", and more recent - Peleg et al. (2017) – "An advanced stochastic weather generator for simulating 2-D high-resolution climate variables".

[Case area] I am missing some climatological information here that will be needed for later understanding of your results. For example, is there any seasonality in the climate? Can you give some examples to what was consider as extreme events in the past (both intensity and duration)? What are the dominant synoptic systems in this region? I guess convective storms during summer and stratiform rainfall during winter? Moreover – high urbanization – where do I see that in Figure 1? "damage potential" – can you give some numbers to quantify the damage?

[3 22] "The post processed data have less than 5 % difference from annual ground truth" – what are the errors for the hourly and daily scales? The study deals with a much finer resolution than the annual scale. More important – how good are the rainfall estimates when extreme events are explored?

[Spatial selection of extreme events] – I understand you are searching for the best strategy to determine what events are 'extreme', but I am not completely understand the rational in your sampling strategies. SS1 – why this particular grid cell was chosen? How different your results be if the single grid cell was elsewhere? SS2 – Why not taking 5 connected grid cells? I am not saying that the strategy is wrong, but it needs to be better explained and discussed.

[5 3] "the five grid cells filled in Figure 2" – that means SS2, right? If so, please indicate this explicitly. Why only from SS2? What with the others sampling strategies? Please explain.

[5 7] "100 randomly selected grid cells" - Why not all grid cells? I guess because of computational time? Are the random selected grid cells represent the terrain? (e.g. all elevations are sampled)

[5 19] and [5 22] "stations" – you mean 'grid cells'?

[Spatial variation] Please revise the text in this paragraph. It is not so clear how you define and compute the spatial variation.

[6 7-9] "The intensity threshold is set to 25% of the maximum 5-minute intensity for the given event with a minimum threshold of 7 mm h$^{-1}$. The areal threshold is set to a minimum coverage of 10 km$^2$" – How did you defined these thresholds? Are you following any physical reasoning? How different the results will be with different thresholds? For example, if 10 mm h$^{-1}$ threshold was applied instead of the 7 mm h$^{-1}$? Some sensitivity analysis might be required here.

[6 9] "Event specific threshold" – what do you mean by that?

[Subsection 3.5.1-3.5.3] Can be shorten. There is no need in so many details, these methods are quite commonly used. I suggest summarizing all statistical methods in one paragraph and supply some key references for the readers.

[9 8-10] "In order to sample events which are meteorologically independent and use the knowledge about extreme events from rain gauge data, sampling method one, only considering one grid cell when sampling extreme events, is selected" – I found this sentence trivial. What I am missing from this paragraph is a clear statement of which sampling strategy to choose, and why. Some discussion might be useful.

[9 18] "convective vs. front events" – this is a good opportunity to discuss some of the differences between convective (more intense, smaller extent, shorter duration) and stratiform (or front) rainfall. It will make the reader better understand the clusters you are suggesting later on.

[Spatial correlation] Interesting. But again some discussion is missing. You can easily compare your results with other studies/climates, see for example: Villarini et al. (2008) – "Rainfall and sampling uncertainties: A rain gauge perspective", Mandapaka and Qin (2013) – "Analysis and Characterization of Probability Distribution and Small-Scale Spatial Variability of Rainfall in Singapore Using a Dense Gauge Network", and Peleg et al. (2013) – "Radar subpixel-scale rainfall variability and uncertainty: lessons learned from observations of a dense rain-gauge network". Are there any differences of the spatial correlation when considering different rainfall types/clusters?

[Table 3] These results are not clear to me. See my comment on the spatial variability above.

[Principal component analysis] I would argue that the PCA results indicate that you cannot point on the most important variables to use for characterization of extreme events. Consider removing the PCA from the paper. I do not see the add value of it for the readers and I think the cluster analysis that comes later is much more important. Unless you want to discuss the cross-correlation or covariance between your variables. In this case you need to put some more effort in the discussion.

[Cluster analysis] Why 4 clusters and not, e.g. 3? I suggest adding some discussion about the climatic systems that force those clusters. For example, higher temperature for the convective

events, deeper atmospheric pressure for the intense convective – something in this direction. I think you need to demonstrate that there is some physical rational behind the clustering.

[12 18] "…only one grid cell…" - Then - why use the weather radar? Use a rain gauge instead. Using a single grid cell you are risking in missing the extreme events completely. Please clarify.

[12 21-22] "This study suggests that further development on a sampling strategy for sampling spatial extreme events is needed" – There are some studies that already goes a bit in a different directions. See recent studies by Lochbihler et al. (2017) – "The spatial extent of rainfall events and its relation to precipitation scaling" and Peleg et al. (2018) – "Intensification of Convective Rain Cells at Warmer Temperatures Observed from High-Resolution Weather Radar Data".

[Title] – 13 years of data is not "long time series" and 1-km is, today, not consider anymore as "very high resolution". I suggest the authors will aim to a simpler title, maybe including weather generator and Germany in the title?

[Figure 1] I suggest merging the two subplots. Can you please also add the locations of the three radars and the distances to the case area?

[Figures 6 and 9] Can go to the supplementary material.

---

## Referee Comment (RC2) · Anonymous Referee #2 · 22 May 2018

Explorative Analysis of Long Time Series of Very High Resolution Spatial Rainfall by E. D. Thomassen, H. J. D. Sørup, M. Scheibel, T. Einfalt and K. Arnbjerg-Nielsen

Recommendation: Reject

I must admit, reviewing this paper was rather challenging. On one hand, the writing is quite good. The structure is clear and the different methods are described and applied in a satisfactory way. The authors seem to know what they are doing. There are no obvious flaws, inconsistencies or statistical fallacies. Still, I strongly advise against the publication of such work (see major comments below). The most important reason for doing so is the fact that the entire paper seems to be a mindless application of different statistical analysis techniques without any clear objective, insight or practical benefit. As the title suggests, the approach is mostly explorative in nature. And while descriptive

studies and exploratory data analysis are important steps in the scientific process, they aren't sufficient on their own without proper context. There are an infinite number of techniques and analyses that you can apply to your data, and an infinite number of features that can be extracted. But what do you actually expect to learn from these analyses? And how do you think this will be useful to design better stochastic weather generators? Maybe the authors already have some good ideas about that. But as long as these are not clearly formulated in the paper, I see no compelling reason to publish this research.

Main reasons for rejecting this paper:

A) Lack of novelty & usefulness: Most of the results and numbers presented in this paper are not interesting or useful. They are just vanity metrics, used to fill tables and give the illusion of hard work. But just because you can extract a lot of information from your data does not mean that this information is useful or actionable. Sometimes, fact-collecting yields nothing more than a collection of facts; no revelation follows. As a result, the paper does not really contribute to the general advancement of our knowledge about extreme rain events. It also does not contain any technical novelty, algorithm or new method that could be applicable to other studies. Therefore, I strongly encourage the authors to think more about what question(s) they actually want to address in this study. The statistical analysis techniques should be selected based on their ability to answer these questions, and not just to fill tables with numbers.

B) No real conclusions: Because of the lack of a real scientific question underlying this work, the conclusions are extremely limited. They can be summarized as follows: (1) A bunch of statistical techniques were applied to analyze features of extreme rain events. Some features are correlated to each other, but not all methods agree on which are the most important ones. (2) Small and large-scale rainfall extremes are not produced by the same physical mechanisms and their statistical properties differ (which has been known for decades). (3) The sampling strategies and event selection method matter a lot, but we still don't know how to properly do this.

C) Questionable link to the design of weather generators: According to the authors, this paper presents the "preliminary steps prior to setting up a weather generator with similar properties as high-resolution radar rainfall data" and "a first step into the direction of good practises to find and analyse single storm events". I strongly disagree with that statement. Actually, I think this study is quite the opposite of good scientific practice. Good practice means you only compute what you really need to improve your understanding and modeling capabilities. This can be guided by a-priori knowledge about how the system behaves or about what end-users need. The paper does none of that, nor does it explain how new weather generators capable of reproducing all the 16 features considered in this study could be designed. Clearly, there is value in trying to use techniques like PCA and cluster analysis to figure out which features are the most important and which can be dropped. But how exactly this is connected and useful for the design of weather generators remains unclear.

Minor Comments:

- Equation 2: What does U represent here?

- Section 3.5.2, p.8, lines 11-17: Maybe you could shorten this section. There are plenty of good references for explaining how clustering works.

- Section 4.2.3: "The spatial variation of extreme events is indicated by calculating the similarity of choice of events by grid cells close to the chosen grid cell." This sentence is not clear. Please reformulate.

- The small size of the study area (38 x 48 km) means that many storm scales and structures will not be properly resolved. Organized systems of thunderstorms can extend several hundreds of kilometers in size. Their properties are thus likely to be misrepresented in the analyses.

---

## Author Comment (AC1) · 3 Jun 2018

We are somewhat surprised and disappointed with the rather harsh review provided by this anonymous reviewer. Let us try to provide an answer to why we think this study is both novel and useful, as well as providing a suitable link to design of weather generators. We understand the reviewer in the way that if these points are justified then the remaining major criticism, lack of conclusions, is also addressed.

We have clearly stated that our key interest lies in identifying properties relevant for a spatial weather generator. The point rainfall generator was originally described by Rodriguez-Iturbe et al (1987) and made operational by Cowpertwait (1994). The main novelty and development between these two key papers is the work on identifying a

suitable number of parameters and how to estimate these, i.e. to obtain a parsimonious description of the variability observed in a historical record. Still, estimation of parameters in the model now known as the Neyman-Scott Rectangular Pulses model is cumbersome and partly subjective due to some over-parametrization of the model, as anyone who has worked with the model knows. We see therefore an identification of dimensions in observed spatial data as a highly needed prerequisite for starting the work on a spatial weather generator. You simply need to know how many variables you should employ in an ideal situation to avoid correlation between variables. This is exactly what is studied in the paper we have submitted and we use as input the longest, most complete, and most accurate precipitation series we know of globally with a resolution needed for urban hydrology applications. As pointed out by both Einfalt et al (2004) and Thorndahl et al (2017) development and deployment of spatial rainfall generators will be of very high value to the urban hydrology community as soon as suitable observational data is available.

However, the reason for publishing such analyses is also, that they can be used outside of the application field that the authors intend to use it for. We can immediately find three other fields of application as discussed briefly below: 1. To validate climate change models of future precipitation in very high resolutions. We point towards the groundbreaking paper by Kendon et al (2014) and their use of high-resolution radar data for verification of precipitation fields at resolutions where no re-analysis data exists for proper validation of model outcomes. 2. To be used for making a typology of extreme rainfall for use in now-casting that better than random projections of the rainfall fields captures realistic temporal evolutions and characteristics as shown in e.g. Olsson et al (2015). 3. To search for good co-variates in e.g. regional modelling of precipitation. The work published in Arnbjerg-Nielsen et al (1996) is a very simple application of some of the tools employed in the present study. This work was the inspiration to identifying the co-variates of the regional model of precipitation extremes developed and published 20 years later by Madsen et al (2017).

We hope that this is sufficient to convince the editor, and perhaps also the reviewer, that our paper is suitable for publication, especially since the reviewer states that the structure of the paper is clear and the methods are applied in a satisfactory way.

References:

Arnbjerg-Nielsen K, Harremoës P, and Spliid H. (1996): Interpretation of regional variation of extreme values of point precipitation in Denmark. Atmospheric Research, 42, (1-4), 99-111.

Cowpertwait, PSP (1994): A generalized point process model for rainfall. Proceedings of the Royal Society of London Series A - Mathematical Physical and Engineering Sciences, 447, 1929, 23-37. DOI: 10.1098/rspa.1994.0126

Einfalt T, Arnbjerg-Nielsen K, Faure D, Jensen NE, Quirmbach M, Vaes G, Vieux B, Golz C. (2004): Towards a Roadmap for use of radar rainfall data in urban drainage. Journal of Hydrology, 299, 2004, 186-202.

Rodriguez-Iturbe I, Cox DR, and Isham V. (1987): Some models for rainfall based on stochastic point processes, Proceedings of the Royal Society of London Series A - Mathematical Physical and Engineering Sciences, 410, 269–288. DOI:10.1098/rspa.1987.0039

Madsen H, Gregersen IB, Rosbjerg D, Arnbjerg-Nielsen K. (2017): Regional frequency analysis of short duration rainfall extremes using gridded daily rainfall data as covariate. Water Science and Technology, 75, 8, 1971-1981. DOI: 10.2166/wst.2017.089

Kendon EJ, Roberts NM, Fowler HJ, Roberts MJ, Chan SC, Senior CA (2014): Heavier summer downpours with climate change revealed by weather forecast resolution model. Nature Climate Change, 4, 7, 570-576. DOI: 10.1038/NCLIMATE2258

Olsson J, Simonsson L, & Ridal M. (2015): Rainfall nowcasting: predictability of short-term extremes in Sweden. Urban Water Journal, 12, 1, 3-13. DOI: 10.1080/1573062X.2015.987428

Thorndahl S, Einfalt T, Willems P, Ellerbæk Nielsen J, ten Veldhuis M-C, Arnbjerg-Nielsen K, Rasmussen MR, and Molnar P. (2017): Weather radar rainfall data in urban hydrology. Hydrology and Earth System Sciences, 21, 1359–1380. DOI:10.5194/hess-21-1359-2017
* * *

---

## Author Comment (AC2) · 13 Jul 2018

Thank you for the thorough review of our manuscript, it is very much appreciated. In the following we will do our best to reply to your comments and suggestions. [page lines]

[2 18] Thank you for providing relevant references. Neither Scopus nor WoS has made the paper by Benoit et al (2018) available yet, but we will include references to Peleg et al (2017) and Peleg and Morin (2014) in the context of highlighting that recent research explores this field. We will likewise identify other suitable domains where the study is relevant, cf. the response to reviewer 2.

[2 24] In line with the comment above, we are happy to include the two suggested

references in the revised manuscript.

[Case area] We will add a short introduction to the climatology in the case area. Regarding the – high urbanisation – it is not shown in Figure 1, but just stated. We believe it is well known that urban areas are vulnerable to highly convective events, which leads to the damage potential. As the paper does not otherwise focus on this we will not go into further details, but we can add some citation stating the observed damages.

[3 22] We will add more information on the radar data product used. The probability for a difference of more than 5 mm between radar data and independent stations (stations not used for radar data adjustment) is 1.4 per station per year. The probability of a difference above 10 mm is 0.1 per station per year (for the methodology, see Einfalt & Frerk, 2011). An analysis of the first 10 years of data shows that there is an underestimation of extreme events for short time steps, which is reduced with growing time intervals (Scheibel & Einfalt, 2015).

[Spatial selection of extreme events] We will elaborate more on this part. SS1 is chosen as the grid cell is in the middle of the catchment and SS1 represent how you would select extreme events from a rain gauge. The chosen grid cell will have great impact on which extreme events you sample; this will be more clear when section 3.3.3 and 4.2.3 and table 3 is explained better. SS2: The five grid cells are chosen to represent a spatial coverage of an urban area, but could also have been connected grid cells. By selecting grid cells with a spatial distance similar to the distances between rain gauges in a city, it is possible to analyse how different events sampled is within a small range, see section 3.3.3.

[5 3] Actually the events are sampled independent of each other with SS1, again to mimic point rainfall measurements. The short distance spatial differences in variation of seasonal occurrence of extreme events are analysed by comparing the result for the five grid cells, which are included in SS2. This will be clarified in the revised version of the paper.

[Figure]

[5 7] Yes, because of the computational time, but also because the correlation between neighbouring cells is clearly very high and calculating that for the entire dataset does not add value matching the extra computational effort.

[5 19] and [5 22] Yes, thank you.

[Spatial variation] We see this comment as a follow up on comment [5 3] and will change the text to make it more clear that the spatial variation in extreme events using SS1 is calculated for the five grid cells from SS2 to analyse the difference in sampled extreme events when changing the grid cell in SS1. The black filled grid cell is used as a reference and compared to the four grey filled grid cells (Figure 2). Extreme events are sampled independent from each of the five grid cells. The number of concurrent events are calculated using Equation 1.

[6 7-9] We agree with the reviewer that the cell tracking procedure could be more advanced. However, we have checked manually that it works well in identifying cells and cell movement and the thresholds applied align well with well-known cell identification and tracking algorithms (Kyznarova and Novak, 2009; Handwerker, 2002; Peleg and Morin, 2012; Dixon and Wiener, 1993). The purpose of introducing thresholds in this paper is to reduce the noise in the tracking, by only tracking rain cells of a certain size and intensity and to compare the importance of this variable to the many other variables use to describe precipitation in the study. Changing thresholds, including performing a sensitivity analysis, would not change the findings of the paper and hence we would prefer not to report the analysis, since it may imply a shift of focus of the paper.

[6 9] Will be corrected to: "The intensity threshold is chosen to be event varying to distinguish…"

[Subsection 3.5.1-3.5.3] We will revise the section and shorten it further, e.g. by referring to the R-packages that we have applied.

[9 8-10] We will revise the sentence to state that the results from Table 2 indicate

that meteorologically independent events are joined when grid cells in a large part of the catchment are considered, which argues that SS3 and SS4 cannot be used. The seasonal distribution of events between SS1 and SS2 does not change and both methods can be considered as sampling strategies for spatial rain events. In order to use the knowledge about extreme events from rain gauge data and compare results SS1 is chosen as the sampling strategy.

[9 18] The two terms will be explained briefly to help the reader understand the clustering later in the article.

[Spatial correlation] Thank you, we will include some of the references above in the discussion of our results.

[Table 3] We hope that the results are easier to understand after we have revised section 3.3.3 as outlined above [Spatial variation]

[Principal component analysis] We agree that the cluster analysis contribute with more interesting results. However, the PCA provides a first result on the amount of information in the data and the cluster analysis builds upon the principal component analysis. Hence we find it important to analyse and report the results of the principal analysis before applying the cluster analysis. We agree with the reviewer that the principal component analysis alone cannot specify the most important variables for characterising events, but it helps specify the number of variables necessary and suggests which variables clusters convective events from frontal events.

[Cluster analysis] We did consider to focus on three clusters, that could be interpreted as: very extreme convective events, convective events and frontal events, i.e. not very different from what discovered with the principal component analysis. What is interesting with four clusters is that the clustering can still be described by well-known weather phenomena. We will include this in our discussion of the results. We haven't included temperature or atmospheric pressure in the analysis, but agree with the reviewer that these variables might be of interest, especially if making simulations of a weather generator by conditioning on the current state of the atmosphere.

[12 18] The events are sampled using one grid cell (SS1) and as shown in 4.2.3 a small change in location of the given grid cells changes the sampled extreme events a lot. But still we find this method the most appropriate to sample independent extreme events. However the analysis of the extreme events sampled are performed on the entire radar image. The characterisation of the extreme events by the 16 variables and the rain cell identification and tracking is not affected by the sampling strategy. The sampling strategy only determines which events are analysed.

[12 21-22] Thank you for pointing us towards these very recent studies, we will relate our results to them. We still think the main conclusion is that the scientific field has not reached consensus on how to characterize precipitation extremes except when based on point measurements.

[Title] We will consider changing the title to reflect the content of the article best possible. When changing the title we will keep in mind that one of the characteristics that make this data set rather unique is that it has been validated and corrected against ground measurement, while many other studies focus on analysis of the outputs from the weather radars without correcting for the shortcomings of this measurement device.

[Figure 1] The location of the radars and the distance to the case area will we added to section 2.2. We agree that the three sub-figures in Figure 1 and 2 to some extend contain redundant information, but found it difficult to read the plot if the two subfigures were merged. We will give it another try.

[Figure 6 and 9] We will consider moving the suggested Figures to the supplementary material if it does not add value to the reader after all the other suggested revisions have been made.

References: Dixon M, Wiener G. 1993. TITAN - Thunderstorm Identification, Tracking, Analysis, And Nowcasting - A Radar-Based Methodology. Journal Of Atmospheric And

Oceanic Technology, 10, 6, 785-797

Einfalt, T., Frerk, I. (2011) On the influence of high quality rain gauge data for radar-based rainfall estimation, Proceedings 12th ICUD, Porto Alegre, 11-15 September.

Einfalt, T., Scheibel, M. (2015) Vergleich von Extremwertstatistiken aus Radarmessungen und Regenschreibern. Hydrologie und Wasserbewirtschaftung, 35.15.

Handwerker J. 2002. Cell tracking with TRACE3D - a new algorithm, Atmospheric Research, 61, 1, 15-34.

Kyznarova and Novak P. 2009. CELLTRACK - Convective cell tracking algorithm and its use for deriving life cycle characteristics. Atmospheric Research, 93, 1-3, 317-327

Peleg N, Morin E. 2012. Convective rain cells: Radar-derived spatiotemporal characteristics and synoptic patterns over the eastern Mediterranean. Journal of Geophysical Research - Atmospheres, 117, D15116